# New Trends in Bio-Based Aerogels

**DOI:** 10.3390/pharmaceutics12050449

**Published:** 2020-05-13

**Authors:** Loredana Elena Nita, Alina Ghilan, Alina Gabriela Rusu, Iordana Neamtu, Aurica P. Chiriac

**Affiliations:** “Petru Poni” Institute of Macromolecular Chemistry, Grigore Ghica Voda Alley 41-A, RO-700487 Iasi, Romania; lnazare@icmpp.ro (L.E.N.); diaconu.alina@icmpp.ro (A.G.); rusu.alina@icmpp.ro (A.G.R.); neamtui@icmpp.ro (I.N.)

**Keywords:** bio-based aerogels, polysaccharides, proteins, bio-applications

## Abstract

(1) Background: The fascinating properties of currently synthesized aerogels associated with the flexible approach of sol-gel chemistry play an important role in the emergence of special biomedical applications. Although it is increasingly known and mentioned, the potential of aerogels in the medical field is not sufficiently explored. Interest in aerogels has increased greatly in recent decades due to their special properties, such as high surface area, excellent thermal and acoustic properties, low density and thermal conductivity, high porosity, flame resistance and humidity, and low refractive index and dielectric constant. On the other hand, high manufacturing costs and poor mechanical strength limit the growth of the market. (2) Results: In this paper, we analyze more than 180 articles from recent literature studies focused on the dynamics of aerogels research to summarize the technologies used in manufacturing and the properties of materials based on natural polymers from renewable sources. Biomedical applications of these bio-based materials are also introduced. (3) Conclusions: Due to their complementary functionalities (bioactivity, biocompatibility, biodegradability, and unique chemistry), bio-based materials provide a vast capability for utilization in the field of interdisciplinary and multidisciplinary scientific research.

## 1. Introduction

The interest in aerogels has been dating for a long time, but products based on such compounds have increased greatly in the last few decades due to their special properties. Aerogels as solid materials present a porous network-like nanostructure, which exhibits special characteristics, such as low density and thermal conductivity, high surface area and degree of porosity, excellent impact damping properties, flame and moisture resistance, low optical index of refraction, and low dielectric constant. The aforementioned properties recommend them for various applications in medical, pharmaceutical, and cosmetics fields, the construction sector, wastewater treatment, environmental pollution, catalysis, and the food industry [1,2,3,4,5,6,7,8,9,10,11,12]. On the other hand, high manufacturing costs and poor mechanical strength are restraining the market growth, with the major challenge in this market being dust contamination in critical end-user industries, such as aerospace or pharmaceuticals. Even so, studies on aerogel market mention a growth at a compound annual growth rate (CAGR) between 30.8% and 31% over the forecast period 2017–2023, while the aerogels for the personal care market, for example, will register a 26.8% CAGR in terms of revenue, whereas the publisher expects the market value to reach a CAGR of around 9% during 2019–2024 [13,14].

Aerogels are materials obtained by replacing the solvent in the meshes of a gel network with air. This replacement is carried out during the drying process, which can be done by supercritical drying, but ambient pressure drying was also attempted. Aerogels are made up of about 95% air or gas by volume and are therefore very light in weight and have high porosity presented in mixtures of mesopores, micropores (<2 nm), and macropores (>50 nm), with an exceptional internal surface area, depending on the reaction conditions [15,16].

Aerogels were first reported by Kistler in 1931 [17]. Although it has been made clear since then that these materials open up new opportunities, the complexity of the obtaining process has restricted their development. The researchers returned to them only after a few decades, when the evolution of the sol-gel method was combined with the supercritical extraction technology. Despite these limitations, a research effort in the development of aerogels was recently registered due to their special characteristics (Figure 1) [18,19,20,21,22]. It should also be emphasized that these defining features have prompted NASA to use silicone-based aerogels for space missions. Moreover, polyamide aerogels and ultra-light cross-linked aerogels (X-Aerogels) have also been researched by NASA because of their distinct properties: These materials have very good mechanical properties, high porosity, and, although they have a very low density, are 100 times stronger [16,23].

Aerogels can be divided into two broad categories, namely inorganic and organic, each category being further divided according to the nature of the materials used in the design of the gel structure. (Figure 2) [25]. Of these, biodegradable and biobased polymers are of increasing pursuit, as the use of these compounds can be an alternative for reducing the impact on the environment [26]. The interest towards these materials is generally presented interdependently and connected with the synthesis routes of the biopolymer aerogels. Their possibilities of functionalization with defined strategies provide them with specific properties in interdependence with their future applications [3]. Thus, polysaccharide-based aerogels have been classified in relation to their applications in environmental engineering, buildings, medical practice, packaging, and electrochemistry, as well as subsequent approaches [11].

Usually, bio-based aerogels are produced from renewable resources, such as sugar cane, proteins, starches, and plant oils. Moreover, biopolymers (such as chitosan, alginate, pectin, lignin, cellulose, protein, etc.) have already been successfully used for the preparation of aerogels, with specific characteristics suitable for biomedical applications, such as tissue engineering, regenerative medicine, and drug delivery systems [27]. As an alternative to the conventional organic solvents used as templates for porous materials, ionic liquids (ILS), also known as “green solvents”, are frequently used in the manufacturing of aerogels based on polysaccharides. ILS are organic salts that are liquids at room temperature, with high thermostability and electric conductivity, and are environmentally-friendly, exhibiting easy recyclability [28]. As the dissolution of some polysaccharides in organic solvent or water is the major disadvantage that can hinder their utilization in obtaining aerogels for medical applications, ILS can provide the most needed processing platforms of these biopolymers as high added-value materials. Moreover, the combination of green solvents, such as ILS, used in the dissolution of polysaccharides, together with the utilization of an environmentally-friendly manufacturing technique, such as supercritical fluid (SCF) technology, which can easily be adapted to extract the ILS, has enabled the development of various porous bio-based aerogels. Thus, polysaccharides, such as chitin [28], cellulose [29,30,31], starch [32], or lignin [33], have been used to obtain aerogels with different degrees of porosity using ILS. Moreover, ILS, such as 1-allyl-3-methylimidazolium chloride (AMIMCl), contribute to the gelation of cellulose, leading to the formation of nanoporous materials with a uniform structure and high degree of flexibility [31]. On the other hand, one of the physical properties of ILS that can influence the surface area of the materials is the melting point. Lopes et al. have highlighted in their study the influence of such a property on preparing porous cellulose-based aerogels. They observed that when ILS with lower melting points, such as 1-ethyl-3-methylimidzolium acetate (EmimAc) or 1-ethyl-3-methylimidazolium diethyl phosphate (EmimDEP), were utilized, aerogels with higher surface areas were formed [34].

Although a lot of reviews have been recently published on the topic of obtaining and applicability of aerogels, this type of literature is beneficial, given the multitude of articles on this subject that appear at an extremely intense rhythm, but also due to the innovative aspects that underlie the obtaining of these materials.

In this paper, about 190 articles from recent literature and studies on the dynamics of aerogel research were reviewed to summarize the technologies used in the manufacture and properties of the materials based on natural polymers from renewable sources. Additionally, the applications in the biomedical field of these bio-based materials were also introduced.

## 2. Methods of Preparation:

Generally, methods for obtaining bio-based aerogels are based on the mixing of precursors, followed by a gelling process and, the most important step, the elimination of the pore filling solvents from the wet gels, without substantial reduction of the volume or compaction of the network. This is usually accomplished by converting the pore filling solvent into a supercritical fluid that is slowly released as a gas. This process allows aerogels to retain the structural form of their wet gel precursors (Figure 3).

In the context of multiple possibilities of aerogel applicability, the need to tailor the aerogel properties to meet application-driven requirements has arisen, for example modulation of the pore structure, surface modification, various coating, and post-treatments, including finding alternative sources for raw materials and precursors [6].

Another direction of investigation was dedicated to the formation of the organic–inorganic hybrid networks in order to improve the mechanical resistance and flexibility of the aerogels. This method was successfully applied to siloxane materials [2,35,36,37,38]

## 3. Drying Procedures for Obtaining Aerogels

An essential step in the process of obtaining an aerogel is the drying procedure, the morphology, porosity, and structural integrity of the final structures, depending entirely on this phase. When conventional drying methods are used, capillary pressure can induce the collapse and cracking of the gel pore structure. For this reason, other drying methods were utilized, such as supercritical drying (using alcohol, acetone, or CO_2_), ambient pressure drying, freezing drying, microwave drying, and vacuum drying [12].

### 3.1. Supercritical Conditions

Supercritical (sc) drying consists of heating the wet gel in a closed container until the temperature and pressure exceed the critical temperature and pressure of the liquid trapped in the pores inside the gel. As a result, the liquid and vapor phase became indistinguishable and no capillary forces appeared. After the release of the gas, followed by the cooling of the material, the aerogel was removed from the autoclave. In sc conditions, the liquid/gas surface tension was 0, because there were no longer liquid/gas interfaces. Drying by scCO_2_ can protect the gel structure and produce materials with a low shrinkage rate, a smaller pore size, and a higher specific surface area [39]. By using this method, the nanometer-scale features and pores were preserved, in some cases leading to thermal conductivities lower than those of air.

A major disadvantage of sc drying is the time-consuming procedure [12]. Moreover, significant quantities of solvents and relatively expensive supercritical gas processing are required, which add costs and a possible environmental impact during the manufacturing stage [40]. Supercritical drying requires specific conditions that differ depending on the solvent used. Thus, for example, when the solvent is water, the required critical temperature (Tc) is about 374 °C and the required critical pressure (Pc) is about 22 Pa. In contrast, ethanol requires a Pc of about 6.3 Pa and a Tc of about 243 °C [41]. Supercritical drying of carrageenan, chitin, alginate, chitosan, cellulose, and agar gels is described, which combine renewable feedstocks with “green” carbon dioxide processing. Berglund used sublimation instead sc by producing an aqueous dispersion of cellulose, and the solvent exchanged was further with *tert*-butanol, the product being then supercritically dried [40]. Compressive moduli of 35–2800 kPa were obtained for flexible aerogels of 0.015–0.105 g/cm^3^ (specific moduli of 2–27 MPa·cm^3^/g) [42,43].

Carbon dioxide under pressure (5 MPa) was also used for ionic cross-linking of amidated pectin [4]. The obtained aerogels were ultra-porous with a low density (up to 0.02 g/cm^3^), had a high specific surface area (350–500 m^2^/g), and had a high volume of pores (3–7 cm^3^/g for pores less than 150 nm).

In the context of using biopolymers for aerogel formation, which includes exploitation of their entire potential, a lot of studies were performed concerning the production times and costs reduction to facilitate the scale-up of aerogel making [44].

At the same time, to reduce the price of aerogels, several strategies for simplifying the manufacturing processes were proposed. In this context, supercritical drying was replaced by freeze-drying or ambient pressure drying as cheaper and more environmentally-friendly strategies [16,45,46,47,48,49,50].

### 3.2. Ambient Pressure Drying

One of the methods implemented for industrial purposes is ambient pressure drying as it is a simple technique that allows energy saving. Thus, organic aerogels were obtained using a sol-gel process followed by solvent exchange with typical solvent (such us acetone or ethanol) and then dried under an ambient pressure condition. However, evaporation of the liquid from the hydrogels under ambient conditions can cause major shrinkage or form solid films without porosity [12].

### 3.3. Freeze-Drying (Lyophilization)

Freeze-drying is a procedure of sublimation of the solid, usually frozen, water from the pores of a wet precursor. In this method, the liquid from the wet gel is first frozen and then eliminated by sublimation at low pressures. The resulting gels, called cryogels, are generally higher in density and lower in surface area as compared to aerogels (porosity of up to 80% and only half of the inner surface of an aerogel). This is mainly due to the development of large ice crystals during the formation of the gel network during the freezing process, which leads to a proliferation of the number of macropores and volume shrinkage. In contrast, compared to the aerogels synthesized by hot-drying or vacuum-drying, cryogels show smaller shrinkage and narrower pore size distribution.

The benefits of the freeze-drying process, which is a simple, more economical and environmentally-friendly process, are derived from the use of water as solvent and the simplicity of the drying process, as well as the possibility of being used for bio-based polymers, such as casein, pectin, alginate, gelatin, hyaluronic acid, and cellulose. The disadvantages of this method are the long processing time, the change in volume when the water is frozen, which can sometimes produce the collapse of aerogels, and high energetical consumption. The networks obtained show the thickness and spacing at the micrometric scale, with better thermal and mechanical properties than those of traditional polymeric foams [39].

### 3.4. Other Methods

(a) Other methods, such as microwave drying, have also been used to obtain aerogels with high surface area and suitable porosity. Aerogels obtained by this procedure show comparable structures, but with more small-sized interconnected macropores, to those prepared by freeze-drying. Moreover, this is a faster technique with promising results [12].

(b) Another method is to cut the jet for the production of spherical biopolymer particles. This method was developed by Preibisch’s group [51]. The researchers used solutions of amidated pectin, sodium alginate, and chitosan of 1–3 wt. % as precursors for particle production by jet cutting. The obtained gel particles were subjected to solvent exchange in ethanol, followed by supercritical drying. Particles of aerogel with a large surface area of 500 m^2^/g were obtained, with good adsorption stability and capacity.

## 4. Bio-Based Aerogels

In the 21st century, a new generation of aerogels based on biomass was developed. These materials have been called bio-aerogels and are generally prepared by (i) dissolving biopolymers (such as polysaccharides or proteins) and (ii) gelling the solution (in some cases this step is omitted), followed by (iii) scCO_2_ drying. Since, in most cases, the polymer solvent is not miscible with CO_2_, a solvent exchange step is required. This stage usually leads to the coagulation of the polysaccharides; however, due to the rigidity of the chains, the polymer does not collapse and results in a tridimensional (3D) network [52].

Polysaccharides are considered to be key elements in the construction of biomaterials for life sciences (for example, food, cosmetics, medicines, and pharmaceuticals). The biocompatibility and biodegradability of these biopolymers, as well as the variety of chemical functionality they possess, makes them promising carriers for drug delivery systems. Polysaccharides are known for their ability to self-assemble or self-order into certain physical structures or forms. In this regard, these biopolymers form gel-like structures in aqueous solutions of certain concentrations. This ability was successfully explored in the formation of aerogels, cryogels, or xerogels. Polysaccharide aerogels (Table 1) are very porous (the porosity varies between 90 and 99%) and lightweight, with a large surface area capable of providing improved drug bioavailability and drug loading capacity [12].

In the context of using polysaccharides for aerogel preparation owing to their special properties (which include biodegradability, biocompatibility, bioactivity, non-toxicity, environmental friendliness, low processing costs, and the presence of multiple functional groups, such as hydroxyl, amino, and carboxylic acid groups on their backbones capable of functionalization), complex aerogels (ChiNC/TCNF/CGG aerogels) were prepared from nano-polysaccharides, namely chitin nanocrystals (ChiNC), 2,2,6,6-tetramethylpiperidinyloxy (TEMPO)-oxidized cellulose nanofibers (TCNF), and cationic guar gum (CGG), by following a facile freeze-drying method and using glutaraldehyde (GA) as a cross-linker [49]. The complex was further modified with methyltrichlorosilane (MTCS) for obtaining a compound with superhydrophobicity/superoleophilicity to be used for oil–water separation. The modified complex aerogel was tested for continuously separate oil from water with the assistance of a vacuum setup, and it maintained its high absorption capacity for 10 cycles.

### 4.1. Cellulose-Based Aerogels

The production of aerogels based on low-cost biomass precursors has recently gained interest both academically and commercially, due to the range of economic and chemical advantages. A variety of raw materials can be used for the manufacture of aerogels. The “oldest” organic aerogels were made of cellulose, along with silica-based aerogels [12]. Although cellulose is the most abundant natural polymer and is already widely used in the industry for various applications, the initial attempts to prepare cellulose aerogels have not been very successful.

At the same time, aerogels prepared using cellulose as a natural renewable and biodegradable polymer have the advantage of being biocompatible, with a large porosity and specific surface area.

These properties allow the use of cellulose aerogels in various fields, such as adsorption and oil/water separation, thermal insulation, and biomedical applications.

#### Type of Cellulose

Lin-Yu Long et al.’s review article mentions the three types of cellulose aerogels: natural cellulose aerogels (nanocell aerogels and bacterial cellulose aerogels), regenerated cellulose aerogels, and cellulose-derived aerogels, along with their potential applicability. Additionally, they evidenced the problems that must be addressed to increase the potential of cellulose aerogels as (i) finding efficient, inexpensive, environmentally-friendly, and non-toxic cellulose solvent systems to improve the dissolution efficiency of cellulose, (ii) the improvement of the cellulose aerogels stability by their physical mixing or chemical modifications, and (iii) the shortening of the production cycle by the sol-gel and solvent exchange processes, as well as improving the gel drying methods [39]. They also mention the advantages when using cellulose and derivatives as precursors for aerogels preparation: (i) it can have beneficial effects on the mechanical properties and an increased affinity to moisture; (ii) the stockpile of cellulose raw material is inexhaustible and can be renewed; (iii) cellulose chains are rich in hydroxyl groups, which helps in intramolecular and intermolecular cross-linking through hydrogen bonds, thus making the aerogels preparation process very simple; and (iv) the improvement of the mechanical strength and structural characteristics of cellulose can be relatively easy to achieve due to its high chemical reactivity, resulting in a large number of derivatives with various functions.

At present, cellulose aerogels are mainly prepared by dissolving and regenerating cellulose in an aqueous or organic solvent, without considerable losses of specific surface area [53,54,55]. The advantages brought by the aerogels obtained from cellulose were mentioned by other authors too, especially their very high impact resistance [25] and their high capacity to be crosslinked [48], which gives the products special properties, such as apparent absorption selectivity toward organic solvent rather than water, for example.

In the last decade, polymer nanofiber-derived aerogels based on, for example, nano-cellulose derivatives, nanochitin, were developed. Nanofibers serve as building blocks to form chemically crosslinked and/or physically entangled 3D networks. The new compounds become promising candidates in many fields of applications due to their excellent elasticity, high specific surface area, ultralow density, and tunable chemical composition. The nano-cellulose derivatives, named in relation to the processing methods as nanofibrillated cellulose, nanocrystalline cellulose, and bacterial cellulose, were used as building blocks for producing robust aerogels, despite the crosslinker absence [56,57].

Studies have shown that by using cellulose nanofibrils (CNFs), the contraction of the aerogel can be significantly reduced, thereby improving the performance of the material [23,58]. Due to their biodegradability, low density, high absorption capacity, and high specific surface area, aerogels based on CNFs are of great interest. Such aerogels have been designed by Mulyadi and collaborators to provide materials with excellent water-absorbing tendency [59] (see Figure 4). The prepared hydrophobic gels exhibited low density (23.2 mg/cm^3^), high porosity (98.5%), good flexibility, and solvent-induced recovery properties. More importantly, CNFs offer environmental and economic advantages, as they are the most abundant renewable resources and their manufacture as aerogels requires neither solvent involvement nor complex processes. More than that, it has been well documented and proved that cellulose aerogels are capable of absorbing a variety of organic solvents.

Regenerated cellulose or cellulose II is obtained by dissolving cellulose I in concentrated alkaline solutions, resulting in cellulose with a silky texture, which makes it widely applicable in the textile industry. The difference between cellulose I and II is based on the crystalline structure, which is mainly changed by the way the hydrogen bonds are organized between the cellulose chains. Cellulose II aerogels are very lightweight with a high specific surface area. The gelation is often omitted because cellulose can form 3D structures during the solvent exchange. Changing the processing conditions and the type of cellulose can turn the morphology and properties of cellulose-based aerogels. [27].

In another investigation, aerogels based on interpenetrated cellulose-silica networks were prepared using molecular diffusion and forced flow induced by pressure differences for the impregnation of the wet coagulated cellulose onto the silica phase (polyethoxydisiloxane) [60]. The method for impregnation based on the forced flow induced by pressure difference determined a decrease in the impregnation times by three orders of magnitude in the case of samples with the same geometry, and the nanostructured silica gel formed in situ inside cellulose matrix revealed a threefold increase in the specific surface area of pores for the nitrogen adsorption, compared with the cellulose aerogel alone. Additionally, the composite aerogels presented lower thermal conductivity than that of cellulose aerogel. This behavior was attributed to the formation of superinsulating mesoporous silica inside cellulose pores, while the composite aerogels were stiffer than each of the reference aerogels.

Seantier’s group prepared bio-composite aerogels based on bleached cellulose fibers (BCFs) and cellulose nanoparticles with various morphological and physicochemical characteristics by a freeze-drying technique [61]. The composite aerogels were characterized and compared with the BCF aerogel, and drastic changes were put in evidence by the morphological investigation, specifically SEM and Atomic Force Microscopy (AFM) techniques, and attributed to the variation of the cellulose nanoparticle properties, namely the aspect ratio, the crystalline index, and the surface charge density. The investigation confirmed the appearance of a new organization structure with pores of nanometric sizes, which determined a decrease of the thermal conductivities from 28 mW·m^−1^·K^−1^ for BCF aerogel to 23 mW·m^−1^·K^−1^ in case of bio-composite aerogel. The increase of the insulation properties for the bio-composite was more pronounced for aerogels with cellulose nanoparticles with a low crystalline index and high surface charge (nanofibrillated cellulose (NFC)-2h). Additionally, a significant improvement of the mechanical properties under compression was registered for bio-composite aerogels generated after self-organization in the network.

### 4.2. Lignin-Based Aerogels

Lignin, the second most abundant biopolymer after cellulose, is an interesting choice for aerogels preparation, because it has a rigid, hyperbranched macromolecular structure, is composed of three different types of phenylpropane units, and has numerous functional groups, such as hydroxyl, ether, methoxy aldehyde, and ester. Lignin is an unused resource, with only 2% of the lignin produced worldwide used to obtain materials. Therefore, finding alternative uses of lignin would be commercially beneficial given the abundance of this biopolymer as a raw material.

Recent studies have shown that lignin is a promising phenolic polymer for aerogel manufacturing [62,63,64,65]. Thus, lignin-based aerogels, which have both high porosity and compressibility, have promising bio-industrial uses for adsorption and sound damping materials. Chen et al. used lignin as a substitute for resorcinol and phenol in the production of aerogels [16,66]. Wang et al. reported the preparation of an aerogel with strong mechanical performance based on lignin and cellulose as a green adhesion agent [67]. Their approach, the direct dissolution in ionic liquids and regeneration in deionized water, determined the micro and nanometric assembly between cellulose and lignin molecules. Room temperature ionic liquids (RTILs), a new class of high solubility solvents for cellulose and lignin, were used for simultaneous or separate dissolution of lignin and cellulose. For example, the researchers used 1-butyl-3-methylimidazolium chloride (BMIMCI), a common RTIL compound. After solvent exchange and freeze-drying, aerogels with improved mechanical properties (the Young module up to 25.1 MPa), high-efficiency adsorption, and excellent thermal insulation were obtained [68].

For the first time, Grishechko et al. synthesized and characterized highly porous organic aerogels based on tannin and lignin. The influence of the ratio between tannin/lignin and (tannin + lignin)/formaldehyde was studied by plotting the phase diagram illustrating the range of compositions in which hydrogels can be obtained. The porosity of the resulting aerogels, dried with scCO_2_, was examined in terms of surface, macro- (pore width >50 nm), meso- (2–50 nm), and microporosity (<2 nm). The influence of the composition on the porosity characteristics has been carefully discussed and supported by electron microscopy studies. It has been shown that the gradual replacement of tannin with lignin changed the pore size distribution [64].

### 4.3. Pectin-Based Aerogels

Major structural polysaccharides include pectins, amorphous, white complex carbohydrates, which are found in ripe fruits and some vegetables and are available as a by-product of fruit juices, sunflower oil, and sugar production. Due to their ability to reduce cholesterol levels in the blood, capture toxic cations (lead and mercury), and eliminate them from the gastrointestinal tract and respiratory organs, pectins have multiple applications in the biomedical field [68]. The ability to form gels naturally, to thicken, and to stabilize easily makes pectin a very interesting carrier in the pharmaceutical and biotechnology industry.

Pectins are anionic polysaccharides made up of linear regions of 1,4-α-d-galacturonosil units, and their methyl esters are interrupted by 1,2-α-rhamnopyranosyl units. The presence of carboxyl units along the backbone allows the formation of hydrogel networks when cations are added. Different cations (Ca^2+^, Sr^2+^, Ba^2+^, Ni^2+^, Cd^2+^, Mg^2+^, and Pb^2+^) were used, of which Ca^2+^ was studied for the food industry. The gels obtained have fragile structures, their resistance depending on the concentration of pectin and the pH range, but also the concentration of calcium ions [4]. The utility of pectin-based aerogels derives from the ability to have tunable physical and mechanical properties by incorporating polyvalent cations (replacing sodium and ammonium ions) that ionically bind to the structures. Thus, aerogels produced from 5% pectin solutions can present compression modules up to 70 kPa, which can be reduced to 2 MPa by the simple addition of Ca^2+^ ions. These mechanical properties can be increased dozens of times by doubling or tripling the concentration of the pectin solution [4].

In recent years, special attention has been paid to formulations containing magnetic nanoparticles suitable for biomedical uses. The synergistic combination of an aerogel and magnetic fillers will provide versatility to the resulting product. A pectin-based aerogel as a biodegradable matrix containing maghemite nanoparticles was developed by García-González et al. [69]. The gels were prepared with two different morphologies (monoliths and cylindrical microspheres) and prepared by a combination of sol-gel and supercritical drying methods. For aerogel microspheres, the sol-gel method was substituted by the gel-emulsion process. The obtained aerogel-based materials were evaluated for their physical, stability, and magnetic properties. The magnetic properties of maghemite have been retained in the final material after processing, which gives a high degree of applicability in the medical field as a drug-targeted carrier.

Veronovski’s group obtained aerogels based on pectin synthesized from citrus fruits or apples for potential pharmaceutical applications. Ionic crosslinking was used to obtain the compounds [70]. Thus, spherical and multi-membrane gels were first formed by the diffusion method using a 0.2 M CaCl_2_ solution as an ionic crosslinker. A very high specific surface area was obtained for this type of aerogel (593 m^2^/g). Pectin gels were subsequently transformed into aerogel by supercritical drying using CO_2_. As the specific surface area is one of the key parameters in drug delivery control, pectin-based multi-membranous aerogels have been further used as drug delivery vehicles.

In another study, the biodegradability of pectin-based aerogels was examined by detecting CO_2_ release for 4 weeks from compost media [26]. The results showed that pectin aerogels have higher biodegradation rates than wheat starch, which is often used as a standard for efficient biodegradation. The addition of multivalent cations or clay surprisingly improved the biodegradation rates.

A method for pectin aerogels preparation was adopted via dissolution-solvent exchange-drying with supercritical CO_2_ [71]. The method allows for correlating the thermal conductivity with aerogel morphology and properties investigation, which led to understanding the connection between thermal superinsulating material and its lowest possible conductivity. To adjust the mechanism of pectin gelling, the concentration of the polymer, the pH of the solution, and the presence of bivalent ions were varied. The study showed the need for a trade-off between density and pore size in order to obtain aerogels with low conductivity values.

### 4.4. Alginate-Based Aerogels

Alginate, a polysaccharide that was discovered by Standford more than a century ago [72], is an organic material extracted from seaweed that contains alpha-l-guluronic acid and β-d-mannonic acid residues, which are linearly linked by a 1,4-glycosidic bond. Alginate has been extensively used in the pharmaceutical, food, textile, and paper processing industries for many years [73,74,75,76,77,78].

The literature mentions different methods of obtaining aerogels based on alginate, and, considering the importance of using these materials in biomedical applications, comparative studies have also been carried out. The solvents used, as well as the drying method of the gel, are key parameters that influence the final characteristics of alginate aerogels. Thus, Rosalía Rodríguez Dorado [78] investigated xerogel, cryogel, or aerogel gel beads obtained from alginates of different weights and molecular concentrations and used different gelation conditions and drying methods (e.g., supercritical drying, frozen drying, and oven drying). The study highlights the stability of the physicochemical properties of alginate aerogels in interdependence with the obtaining and storage conditions. Aerogels based on alginate biopolymer are characterized by unique properties, such as large surface, open porosity, good compatibility, and biodegradability [79]. In addition, they have a very low thermal conductivity, a high potential for insulation applications, and a good flexibility and are classified as viscoplastic materials [3]. By adding Ca^2+^ or Al^3+^ ions, aerogels based on alginate with mechanical properties better than those made of pectin can be obtained.

To produce aerogel micro-particles in a larger amount, a continuous emulsion-gelation process was proposed, which was able to produce gel micro-particles in sizeable quantities; a method that demonstrated the industrial relevance [80]. For this approach, the alginate-paraffin oil-Span80 system was taken as a model gelling system and two gelation mechanisms were demonstrated. The method has in view pumping together of an alginate solution and oil through a progressive cavity pump, which were further fed to a colloid mill to produce alginate in oil emulsion in one pass. The emulsion was further gelled by in situ crosslinking with a gelation trigger (acetic acid or calcium chloride), and the obtained gel microparticles were separated by sedimentation or centrifugation and partitioning into an ethanol solution. After solvent exchange to ethanol and supercritical drying with CO_2_, the obtained aerogel microparticles presented a large specific surface area and dimensions in interdependence with the size of the emulsion droplets.

Ahmad et al. studied the possibility of the formation of a non-woven composite based on an alginate aerogel with increased thermal behavior [81]. The researchers soaked a needle punched polyester (PET) nonwoven with an alginate solution system, further immersed in an aqueous CaCl_2_ solution when resulted in a gel inside the PET nonwoven. Alginate-based materials have high hydrophilicity and insufficient mechanical properties. To improve the mechanical properties and extend the range of applications, some approaches have been developed through additional crosslinking [13] or by incorporating reinforcement components. Recently, Martins et al. presented a study on obtaining hybrid aerogels based on alginate and lignin. The new compounds were not cytotoxic and had good cell adhesion [82]. At the same time, they explored the development of hybrid aerogels based on alginate and starch as a second component and evaluated these materials as 3D constructs for bio-applications.

The control of the proanthocyanidins release, including generating antioxidant character, to an alginate aerogel, and even the increase of its mechanical properties, was realized by improving the aerogel composition with pectin and crosslinking with divalent cation (Ca^2+^) and sol-gel method, followed by the freeze-drying process [83]. The new aerogel system presented a good controlled release described by first order and Korsmeyer–Peppas models, while the radical scavenging activity indicated stronger antioxidant activity for the encapsulated aerogel microspheres with higher pectin content.

In order to evaluate the mechanical properties of alginate aerogels, a novel mesoscale modelling approach was recently proposed [84]. In this model, the porous structure of the aerogels was represented at the mesoscale level as a set of solid particles connected by solid bonds. Thus, an elastic-plastic functional model was developed to describe the rheological behavior of alginate aerogels with varying degrees of cross-linking, namely calcium content.

The development of aerogels based on alginates as carriers of plant active substances was presented by Mustapa’s group as one of the innovative techniques in the pharmaceutical industry for improving the solubility and bioavailability of plants [79]. In their study, medicinal herbal extracts were impregnated in alginate aerogels by liquid and supercritical mediums during the supercritical drying process. The alginate aerogels were prepared using CaCO_3_ as the cross-linking compound. The hydrogels were then transferred into molds and stored in the refrigerator (4 °C) until they were completely gelled. Prior to supercritical drying, the hydrogel underwent a successive exchange of solvents (30%, 50%, 70%, and 90% *v/v* in 24 h for each concentration and finished by washing twice with pure ethanol) to remove water and impurities. Alginate alcohol gels were then dried to obtain aerogels.

### 4.5. Starch-Based Aerogels

Starch is a promising source for aerogel formation, due to its low costs and biodegradability. Among starch sources, wheat starch has the greatest potential to form hydrogels with a three-dimensional network, providing many opportunities for pharmaceutical applications as a bioactive carrier [85,86].

In 1995, Glenn and Irving prepared the first starch-based aerogels, referred to as “microcellular foam” [119]. Starch obtained from mixtures of wheat and corn with high amylose content obtained from solutions of 8% by weight was used to obtain aerogels with a low thermal conductivity of 0.024 W/mK. Glenn’s group and Te Wierik et al. independently demonstrated that the low surface area of native starch could be expanded to generate xerogel materials with a specific surface between 25 and 145 m^2^/g^−1^, depending on the method of preparation [12,119,120,121].

A few years later, starch-based aerogels were obtained by the dissolution-retrogradation-solvent exchange method and their use as a matrix for drug administration applications was suggested [87]. Thus, increasing the starch concentration from 5% to 15% led to an increase in density, for example from 0.12 to 0.23 g/cm^3^ for pea starch aerogels [88] and from 0.04 to 0.015 g/cm^3^ for wheat starch aerogels, respectively [89]. A detailed study on the influence of the processing parameters on the properties of starch aerogel was presented by Budtova et al. [90], also emphasizing the importance of processing time for decreasing the specific surface and for increasing the mechanical properties and thermal conductivity. Starch mixtures were prepared by dissolution in water (thermo-mechanical treatment), retrogradation, solvent exchange, and scCO_2_ drying. Amylose starch content ranged from 0% to 100%. Specific surface area, aerogel density, morphology, thermal conductivity, and mechanical properties under pressure were investigated. Finally, a comparison of the thermal conductivity of the starch aerogel with that of other bio-aerogels was made [86].

The group of Ciftci also reported the preparation of starch-based aerogels by using scCO_2_ drying [87]. They investigated the conditions for optimizing the formation of aerogel (the largest surface area and the smallest pore size) by using different processing parameters.

A robust gel with improved mechanical properties based on sodium montmorillonite starch (Na-MMT) was developed by using the freeze-drying method. Glutaraldehyde (GA) was used as a crosslinker in the presence of irradiation [88]. It was found that the microstructure and mechanical properties of chemically cross-linked starch aerogels changed with GA concentration. Na-MMT clay played the reinforcement role. Moreover, the addition of clay created more porous structures and therefore reduced the thermal conductivity of the aerogels. At the same time, high biodegradability of starch/clay aerogels was demonstrated.

Recently, Kenar and collaborators reported the obtaining and properties of some inclusion complexes prepared by mixing starch with sodium palmitate [89]. These complexes possessed polyelectrolytic properties due to the anionic loading of sodium palmitate. When acid was added to lower the pH, these stable dispersions turned into hydrogels. Depending on the chosen drying process, the corresponding xerogels, cryogels, and aerogel were obtained. Solvent exchange with ethanol and scCO_2_ drying of these gels maintained their structure and generated materials with a macroporous internal structure. Starch aerogels with densities between 0.120 and 0.185 g/cm^3^ and BET (Brunauer–Emmett–Teller) surfaces ranging from 313 to 362 m^2^g^−1^ were obtained. The corresponding xerogels and cryogels had weaker properties. This technique provides an alternative way of preparing starch aerogels and eliminates the numerous difficulties associated with starch gelatinization and retrogradation procedures that are currently used to prepare starch gels [90].

In another study, sterile maize starch aerogels with a macroporosity of 1–2 µm, but also with a complex nano-architecture, were prepared through a new methodology that consisted of the addition of zein as a biocompatible porogen [91]. Starch-based aerogels showed good biocompatibility with increased cell viability (>80%).

### 4.6. Chitosan-Based Aerogels

Chitosan is one of the most abundant polysaccharides, along with cellulose, and can be extracted from shells of crustaceans and mollusks, such as shrimps, crabs, and squids. This biopolymer allows various modifications on the backbone, including the deacetylation process, for the further numerous applications in the production of biomaterials, drug administration systems, and as a support for cells and enzymes, etc. [92,93,94,95,96].

Chitosan hydrogels can be formed either by chemical or physical crosslinking. The choice of crosslinking is related to the requirements in applications, namely the stability of the aerogel and the required porosity and homogeneity. After the exchange of solvents with alcohol, chitosan-based hydrogels are supercritically dried to obtain aerogels with high porosity, high surface area, and low density; characteristics that depend on the concentration of chitin and the used alcoholic solvent. In general, larger-area aerogels are synthesized from chitosan obtained by deacetylation of the chitin from crab shell [12].

Rinki et al. prepared chitosan-based aerogels using scCO_2_ and investigated their biological properties [97]. The use of supercritical liquids offers solvent-free, natural, and safe products for biomedical applications. The prepared aerogels showed a large surface area, the nature of the solvent affecting the porous structure of the material. Moreover, the proven antibacterial activity of the prepared materials may be valuable in medicine.

Natural compounds have attracted attention due to their unique properties, but the significant contraction of the aerogel from biomass and from the wet gel to aerogel remains a challenge. For this reason, realizing hybrid aerogels by the introduction of synthetic polymers, such as linear polyvinyl alcohol chains (PVA), in aerogel synthesis has been proposed to form a strong architecture for the gel network. Thus, using supramolecular interaction and covalent crosslinking, an aerogel with good thermal insulation and compressible properties based on chitosan and PVA was obtained. It has also been shown that the addition of PVA can cause the desired orientation shrinkage and linear elasticity at low pressure, with respect to the chitosan aerogel [98,99,100].

Aerogels that combine the characteristics of the nanostructured porous materials, i.e., the extended specific surface and porosity at the nanoscale, with the remarkable functional properties of chitosan, were obtained from solutions of the biopolymer in ionic liquids [100]. The effect of the solvent was studied by using 1-butyl-3-methylimidazolium acetate and 1-ethyl-3-methylimidazolium acetate. The process of obtaining aerogels had three stages: (1) the formation of physical gels by diffusion of anti-solvent vapour and (2) liquid phase exchange, followed by (3) scCO_2_ drying. The structural characteristics of the resulted chitosan aerogels were distinctive and could be related to the initial solvation dynamics. The obtained aerogels have increased potential for use in the pharmaceutical industry as materials for encapsulation, retention, and transport of drug molecules with chitosan affinity.

A new organometallic compound was obtained by complexing Au (III) to chitosan aerogels in the presence of an Au (III) chelating agent (dimercaprole) [101]. A material with good catalytic activity was obtained in the oxidation reaction of aliphatic alcohols, benzyl alcohol, and ethylbenzene.

New pH-sensitive biodegradable aerogels based on chitosan and polypropylene glycol with applicability in the biomedical field were developed [102]. Microwave irradiation using organic acids and propylene glycol as crosslinkers, followed by their transformation into porous biomaterials through the lyophilization process, was used as the method of preparation. Biodegradability, bioactivity, and pH response were analyzed. An anticancer drug release profile was investigated showing promising results for applicability.

Investigations were made for improving the mechanical properties of chitosan aerogels. Extracted graphene oxide was introduced as fillers into chitosan aerogels [103]. The porosity of the new composite aerogels was realized by an environmentally-friendly freeze-drying process with various content of graphene oxide (0, 0.5, 1.0, and 1.5, wt. %). It was concluded that the microstructure of the fillers was developed in the network structure. The porosity of the new aerogels was as high as 87.6%, and the tensile strength of the films increased from 6.60 to 10.56 MPa with the recombination of graphene oxide. Additionally, the crystallinity of the composite aerogels increased from 27% to 81%, most likely due to the chemical crosslinking of chitosan by graphene oxide, thus improving the mechanical properties.

### 4.7. Protein-Based Aerogels

Natural protein-based aerogels open up new opportunities in the biomedical field due to their biodegradability and biocompatibility. These types of biomaterials have already been found useful in controlled drug delivery [104].

Recently, Mallepally’s group [105] manufactured protein-based gels from silk fibrous sources, with the potential to be applied as scaffolds in the field of tissue engineering. These gels were subsequently processed into aerogels with improved mechanical and textural properties (high specific area, high porosity, and interconnected porous network) and showed good cytocompatibility and cell adhesion.

Aerogels composed of soy proteins and nano-fibrillar cellulose were also obtained (about 70% soybean load) [106]. The resultant aerogel composites had a high compressive strength of up to 4 MPa and were less prone to structural damage upon contact with a polar/non-polar solvent [7].

In order to predict and control the properties of protein-based aerogels, it is important to identify the relevant parameters that influence the precursor protein hydrogel and to correlate them with the resulting aerogel characteristics. Hence, extensive research has focused on the conditions used during egg white gel formation under temperature [107]. The pH and ionic strength were identified as decisive factors in manipulating the hydrogel structure made of egg proteins. The purpose was to characterize different protein-based hydrogels in terms of their rheological and textural properties and to correlate these properties with the final properties of the resulting aerogel. It was examined whether explicit properties of aerogel can be obtained, depending on the type of protein and the mechanism of hydrogel formation.

The idea was that the differences that appear in the properties of the hydrogels translate into different characteristics of the aerogel, especially in terms of adsorption capacity and loading capacity. Whey and egg white proteins and sodium caseinate were chosen as protein sources for gel formation due to their unique ability to form hydrogels. They were characterized in relation to the hydrogel properties and compared to the resulting aerogels [108].

#### 4.7.1. Albumin-Based Aerogels

Another protein used more often in the preparation of aerogels is albumin due to its use in biomedical applications and its ability to self-assemble in the presence of other polymers, forming networks [109,110,111,112]. Li et al. used mixtures of albumin, camphor, and formaldehyde to develop an aerogel structure [113]. In another study, aimed at replacing camphor with other additives, such as tetra-amines and several tannins, complex materials were designed using different preparation protocols (variations in concentration, pH, curing process, type of additives, and the nature of tannins) [114]. Low density porous, flexible foams were obtained [115].

#### 4.7.2. Casein-Based Aerogels

Casein is a phosphoprotein that can be separated into different electrophoretic fractions, such as α-casein, κ-casein, β-casein, and γ-casein, wherein each component differs in the primary, secondary, and tertiary structure, the amino acid composition, and molecular weight. Casein also includes amino groups, ketones, and hydrazine groups [116]. Casein is a milk protein, which is an interesting example in the sense that it can be cross-linked enzymatically or chemically to form aerogels but remains highly biodegradable. A recent approach uses bivalent and trivalent cations as crosslinkers for a renewable pectin and sodium montmorillonite clay system, which caused the supramolecular chains to bind and led to the modification of the mechanical properties [26]. This kind of benign crosslinking method is more economical and durable than the use of glycoaldehyde, diisocyanates, and other highly reactive and toxic chemicals. Another non-polluting process is the preparation of casein/clay (sodium montmorillonite) dried by lyophilization. Very low densities (0.07–0.12 g/cm^3^) and good compressive properties (90–5600 kPa) were registered for the newly prepared aerogels.

#### 4.7.3. Gelatin-Based Aerogels

Gelatin is a widespread biobased protein that is obtained from collagen, mainly extracted from bones, tendons, and skin. Due to its biodegradability, biocompatibility, and non-immunogenicity, it is widely used in the pharmaceutical industry. The hydroxyl, carboxyl, and amine groups on the gelatin chains make them easy to dissolve in water and subsequently form a heat-reversible physical gel at a relatively low temperature, in which the macromolecular chains recover the triple-helix structure of collagen [117].

Gelatin aerogels were successfully obtained by crosslinking with formaldehyde and coating the surface of siloxane aerogels by thermal chemical vapor deposition (CVD) of methyltrichlorosilane. The resulting materials had a low density (5–8 kg/m^3^) and high porosity (>95%) with uniform pore size [118].

## 5. Application in the Biomedical Field

The use of aerogels in the field of biomaterials is relatively recent. In the last decade, aerogels have attracted interest in the biomaterials community due to their special properties (large porosity, high internal surfaces, controlled pore diameter, and 3D interconnected structure). Biobased aerogels additionally provide superior cytocompatibility, biocompatibility, and biodegradability and can be used successfully in biomedical applications, such as tissue engineering [81,122,123,124,125,126,127], reservoir drug delivery systems [128], biomedical implantable devices (pacemakers, stents, and artificial heart valves), disease diagnosis [129,130], and antibacterial materials [131,132] etc. [133,134,135,136,137,138,139,140] (Figure 5, Table 2).

Aerogel materials prompted a growing interest in pharmaceutical sciences for drug delivery applications owing to their high surface areas, high porosity, open-pore structures, increased bioavailability for low solubility drugs, improved stability, and release kinetics [133].

These properties can also be tuned and controlled by manipulating the synthesis conditions, with nanostructured aerogels representing a promising class of materials for the delivery of various drugs, as well as enzymes and proteins, as already presented in the specific literature and specialty reviews [141,142,143].

### 5.1. Aerogel in Drug Delivery

In the last few years, we have witnessed an increase in the number of studies on obtaining new systems that can be used for controlled drug delivery through different routes of administration. One of the major challenges imposed by these applications was to obtain formulations with precise spatio-temporal control of drug delivery, but which also has a protective role regarding the degrading effects of the physiological environment on the drug before reaching the target site. Another challenge was to improve the bioavailability of drugs with low solubility through the use of aerogels that allow the drug to disperse into the porous substrate [98,100,133].

Drugs can be incorporated into aerogels by two methods. In the first case, which is considered the simplest, allowing us to effectively incorporate a wide range of therapeutics with specific criteria (solubility and dispersion in the soil phase and chemical stability at variations of pH and temperature, etc.), the drug is added in situ during the gelling process or the aging process. The second modality consists of the addition of the drug, ex situ, by absorption or precipitation in the dry aerogel. By this method, the drug in a liquid or gaseous phase [150] is incorporated into the aerogel matrix. But this method has some restrictions, more specifically the slow diffusion capacity of the drug through the pores of the matrix. Thus, the interactions between the drug and the aerogel will be influenced by the ion charges and molecular size of the drug, as well as by the chemical functionality and biodegradation capacity of the aerogel, and which will consequently affect the rate of drug release from the matrix. Additionally, depending on the nature of the aerogel, as well as the design manner of the aerogels (as a multi-membrane reservoir, hybrid, magnetic aerogel, etc.) and the drug used, the release profile can be modulated [143,144,145,146,150,152].

Over the years, a large variety of aerogels from different sources, such as proteins, polysaccharides, or hybrids (for example chitosan or gelatin combined with silica), were used [7,68,147,148,151,153,155] in the administration of drugs with various biomedical applications. Moreover, the utilization of different polysaccharides (alginates, chitosan, pectin, etc.) in the preparation of aerogels have sparked a lot of interest as they have demonstrated the capacity to modulate not only the mechanical properties but also the degree of swelling and the adhesion properties. Therefore, polysaccharide-based aerogels with wound dressings applicability can be obtained, which can rapidly absorb the exudate from the wound level with the simultaneous release of the drug previously incorporated in the matrix.

Due to their improved stability, availability, low toxicity, and cost, Mehling et al. [74] produced aerogels by using different polysaccharide precursors (specifically potato and modified starch alginate), which were then loaded with ibuprofen and paracetamol. They showed that the properties of the matrix and its structural characteristics have a major influence upon the drug loading and release kinetics. Thus, the study results indicated that the amount of drug loaded in this type of system is gradually increasing with the rise of the specific surface area, the latter being related to the average pore size.

Therefore, in the case of aerogels, the high porosity and the capillary forces enable a significantly large loading capability of the drug inside the matrix. In contrast, the rate of drug delivery is influenced mainly by the strength of the aerogel matrix in water and the crystallinity of the drug. Synthesis of aerogels based on silk fiber proteins and with applications in drug administration was described by Marin et al. [155]. Ibuprofen loaded in the aerogel matrices was in an amorphous form, probably due to interactions with fibroin. The in vitro release of ibuprofen was performed in two different stages: a rapid phase in which over 75% of the weight of ibuprofen was released in the first 100 min and a slower stage lasting from 100 to 360 min, in which about 15% of ibuprofen was released. The further 10% was roughly connected to the aerogel matrix and could only be released after its degradation.

Biodegradable and pH-sensitive chitosan aerogels for biomedical applications were prepared by an eco-friendly method that involved the dissolution of chitosan flakes in acetic acid solution, heating, and cross-linking. The drug was incorporated after the dissolution process was completed. Following the obtaining of a homogeneous solution, the samples were irradiated using 400 W for 20 min and freeze-dried, resulting in a very porous material. Bio-tolerant acids and propylene glycol were used as cross-linking agents. Through this innovative approach of simultaneously utilizing two different cross-linkers to obtain 3D structures, very light materials with higher porosity, and pores with different dimensions, shapes and distributions were synthesized. The applicability of these chitosan-based biomaterials as controlled drug delivery systems for anti-cancer drugs was investigated, and their stimuli-responsive behavior was also analyzed in response to the changes produced in the physiological environment (pH and polarity) [102].

Among the various polysaccharide-based aerogels (such as starch, alginate, and pectin) that have been evaluated as drug carriers in different pharmaceutical formulations, using ketoprofen and benzoic acid as model drugs, whey protein-based systems have demonstrated increased capacity for anti-inflammatory drug loading. After incorporation of ketoprofen, the release mechanism was investigated in different media, simulating gastrointestinal conditions (gastric: pH = 1.2 and intestinal: pH = 6.8 fluid), and the aerogels showed a sustained behavior [104].

Surfaces and interfaces outlined an edge between a material and its nearby environment and they were conductive media for chemical and biological processes. Thus, in the case of the drug loading process, the specific surface chemistry that mediated drug-aerogel matrix interactions had a great impact on the embedding capacity of the materials. For example, the release of benzoic acid from aerogel microspheres was more rapidly compared with ketoprofen, due to the size difference between the two drugs. Because benzoic acid is smaller, it diffuses more easily through the polymer-based network [149].

García-González and Smirnova [147] prepared new nanoporous aerogels based on starch with potential biomedical applications. The aerogels synthesized by them exhibited large areas (ranging from 100 to 240 m^2^/g), low density (0.1–0.25 g/cm^3^), and high porosity (85–90%). These aerogels loaded with ketoprofen were tested as drug carriers. The therapeutic release mechanism followed a two-step process with a rapid dissolution of 56% by weight of the total amount in the first step. Ketoprofen strongly interacted with the aerogel matrix and therefore could not be diffused until degradation of the matrix (erosion) occurred [147].

Additionally, multi-membrane aerogels based on pectin incorporating theophylline and nicotinic acid were successfully prepared and used as drug delivery reservoirs. Following the drug release study, the pectin-based aerogels have shown a controlled release rate with a high release rate (very close to 100%) [75].

In another study, pectin-based aerogels were loaded with nicotinic acid as a bioactive compound, a stage that represented the first step in the sol-gel process, utilized for aerogel synthesis. The obtained pectin systems in the shape of microspheres were coated with several layers, and the release mechanism of nicotinic acid embedded in the matrix revealed a strong dependence on the number of layers deposited on the surface of the aerogel microspheres. It was found that the triple membrane aerogel could not control the release. Moreover, a supplementation of the number of deposited layers (at 5) resulted in a controlled delivery of the drug [154,183]

Recently, Zhao et al. [151] obtained an aerogel utilizing polyethyleneimine, grafted with cellulose nanofibers by the freeze-drying technique, and evaluated their potential applicability as versatile drug delivery vehicles. The morphology and the structure of the materials characterized by SEM, FTIR, and XPS, together with the drug delivery studies, revealed that the aerogels can incorporate and release in a controller manner sodium salicylate, a therapeutic agent used in diseases such as diabetes, arthritis, and cancer treatment.

### 5.2. Aerogel for Tissue Engineering

Over the last few decades, the lack of donors, the negative immune response, and infection are the principal limitations that have hindered the utilization of tissues transplanted from both humans and animals for tissue engineering applications. In order to avoid such situations, this research domain has focused on finding alternative approaches for regeneration, reconstruction, or replacement of affected tissues through biobased materials [156,158,159,160].

Thus, as any biomaterial used as a tissue engineering scaffold, the aerogels require particular characteristics, such as suitable microstructures with interconnected pores with dimensions that allow the integration and vascularization of tissues and biocompatibility. Materials must also have adequate surface chemistry to allow cells to attach, proliferate, differentiate, and finally create a new extracellular matrix, as well as a controlled degradation rate and tailored mechanical strength with the ability to be processed in a variety of shapes and dimensions [161].

Due to its architecture characteristics (high open porosity-mesoporosity comparable to the native extracellular matrix, open structure, large specific surface area, and large pore volume), aerogels have recently been proposed for tissue engineering applications and as therapeutic/regenerative platforms for the delivery of specific drugs (the latter was previously discussed in a distinct section above). Nevertheless, the absence of macroporosity in these structures restrict their use for tissue engineering purposes and prompts research on finding some approaches to better stimulate the cell migration in the matrix network [162,163].

Therefore, Silva et al. synthesized chitin-based aerogels as macroporous scaffolds for tissue engineering applications [159], chitin being modified to obtain a bioactive material. In addition, the aerogel preparation technique was improved to increase the porosity in the matrices. Aerogels based on chitosan with applications in tissue engineering were also obtained and characterized [121].

In another study, a new approach to obtain hybrid aerogels was proposed by Quraishi et al., starting from algae and lignin and using CO_2_-induced gelation by solvent exchange, followed by supercritical drying [160]. The aerogels were tested in vitro and in vivo, showing a lack of toxicity and good cell adhesion, properties that are encouraging for their use in tissue engineering [160].

Another aerogel was obtained by Lu et al. [161] by crosslinking a mixture of nanocellulose and collagen with a dialdehyde derivative. A biocompatible composite, stable in the physiological environment, with good physical and microstructural properties, was obtained. Aerogel formation has been shown to occur by embedding collagen in the matrix produced by cellulose dialdehyde fibers. In addition, the composite aerogel had good biocompatibility and was non-cytotoxic, highlighting the great potential of such a system to be used as a tissue engineering scaffold [162].

Engineering of the bone tissue is another area where aerogels have found promising utility due to their special properties [163]. However, aerogels do not have suitable mechanical properties to withstand the strength required for bone tissue engineering applications.

In this regard, efforts have been made to obtain composite aerogels with improved mechanical properties by reinforcing with different mixtures, such as biopolymers or inorganic fillers [163]. The use of mixtures, such as lignin, confers enhanced mechanical properties, increasing the osteoconductivity or cell adhesion [7,163].

Another proposed material for bone scaffolding was based on silica gel and poly-ε-caprolactone. Starting from the premise that poly-ε-caprolactone (PLC) is a biocompatible material, a polyester frequently used in biomedical applications, Ge et al. [163] designed a bone substitute that allows seeding of bone cells. The authors evaluated the influence of embedding silica gel on the applicability of the final product. The biocompatibility evaluation of this compound indicated that the presence of the silica in the macroporous matrix prevented any cytotoxic effects induced by the PLC membrane during extended periods of tissue culture and, consequently, improved cell survival.

Cardiovascular diseases are the leading cause of death in the world, which generates a high demand for biomaterials with proper biological characteristics for cardiovascular implantable devices. However, one of the biggest failures of implantable biomaterial is the initiation of thrombosis formation when the materials come in contact with blood. Therefore, for an aerogel to be applied as a cardiovascular implantable device (e.g., valves), besides the specific biomechanical properties, it needs to fulfill a series of requirements, including low inertia, biocompatibility, and hemocompatibility to avoid deposition or adsorption of plasma proteins onto the surface, which may trigger acute immune response [148]. These characteristics are important in maintaining the durability of the valve. Yin et al. studied the use in cardiovascular applications of a macroporous aerogel based on polyurea-nanoencapsulated with silica gel, and the materials demonstrated good hemocompatibility [157]. Additionally, no changes in normal platelet function were found and no acute immune response was observed in blood plasma after exposure, which makes them promising engineering approaches for heart valve replacement [184].

### 5.3. Aerogel for Biomolecules Immobilization

The fact that aerogels possess unique physicochemical characteristics, biocompatibility, and mechanical robustness has led to the encapsulation and immobilization of various biomolecules into aerogels matrix, thus obtaining a new type of bioactive scaffold, which has found applicability in medicine [164,165]. The biomolecules can be encapsulated either in situ or after supercritical drying, or they can be encapsulated in the wet gel during the sol-gel reaction, followed by the supercritical drying process. An intensely studied biomolecule, as an enzyme catalyst that can be incorporated into the nanostructure of an aerogel, is lipase. It has been found that incorporation of lipase into the structure of an aerogel maintains its enzymatic activity, and, in some cases, it has even been improved [158].

But unfortunately, this still remains a subject that needs more research as the immobilization of proteins in aerogels by using the harsh conditions of the supercritical fluid (SCF) technique can lead to the incapacity of most biomolecules to keep their structure undamaged within the gels during the processing.

### 5.4. Aerogel for Wound Care

The last important biomedical field where aerogels have been applied is wound healing; it is a complex and dynamic process that lasts various days and weeks and, which, through several stages, enables skin restoration after being injured. Moreover, it involves a cumulative and controlled activity of inflammatory, vascular, connective tissue and epithelial cells from the moment when the injury is inflected until the wounds are healed.

Given these issues, an ideal wound dressing should maintain a moist environment at the wound interface, act as a barrier to microorganisms, allow gas exchange, and remove excess exudates. In addition, as any material that comes into contact with the body, they must be non-toxic, biodegradable, biocompatible, non-adherent, and easy to remove after use [166,167,168,170]. It is also desirable for the synthesized material to bring several additional beneficial properties, such as antimicrobial properties, to be loaded with substances that induce wound healing (providing growth factors, cells, and different types of drugs) [169]. The use of aerogels for wound healing allows the formation of a wet gel at the interface between the wound and the material without destroying the proper hemodynamic balance and thus avoiding the traumatic removal of the dressing from the injured skin normally induced by conventional products.

Between the materials successfully used in wound healing, those based on polysaccharides occupy a special place because they are biocompatible, largely biodegradable, and often have a high-water absorption capacity [171,174].

Among them, chitin and chitosan-based aerogels occupy a special place due to their special properties (antifungal and bactericidal character, high permeability to oxygen, stimulating fibroblast proliferation). In order to treat chronic wounds and prevent the subsequent contamination with pathogen agents, vancomycin, an antimicrobial drug, was loaded into chitosan aerogel beads manufactured through the scCO_2_ technique. The high porosity (>96%) and the large surface area (>200 m^2^/g) promoted a fast release of the drug, highly necessary at the wound site in order to prevent the spreading of microbes shortly after debridement. In addition, the in vitro cytocompatibility on fibroblasts evidenced no harmful effect on cells, highlighting the potential of these chitosan-based aerogels as dressings in the management of chronic wounds [178].

Another polysaccharide successfully used as a wound dressing is nanocellulose that is non-toxic, non-allergic, and biocompatible. Moreover, nanocellulose can endow the dressings with the capacity to absorb and retain moisture at the wound site, while having many advantageous effects in the wound healing process, such as minimizing inflammatory response and stimulating fibroblast proliferation [175,176]. Although nanocellulose-based aerogels possess many unique features, such as nontoxicity nature, high porosity, and excellent mechanical properties, they lack antimicrobial properties and stimuli-sensitive character, which limit their utilization in the wound care field. Through the embedment in their 3D network of inorganic nanoparticles or conductive agents, those properties were improved. In this regard, Hosseini et al. synthesized ternary nanocomposite aerogels based on bacterial cellulose, embedding not only silver nanoparticles (AgNP) as antimicrobial agents, but also polyaniline (PANI) particles with rose-like morphology formed in situ within am aerogel network for tailored porosity. All aerogels had higher porosity (>80%) and great elastic properties suitable for a wound dressing. As can be seen in Figure 6, although the cell attachment was not promoted onto nanocomposite aerogels, cell proliferation recorded an on-going increase throughout the incubation period [172]. Another versatile composite aerogel based on nanofibrillated cellulose with copper-containing mesoporous bioactive glass (Cu-MBG) inclusions was successfully obtained by Wang et al. [177]. The reinforcement of the nanocellulose matrix with Cu-MBG particles enhanced the aerogel’s capacity of water absorption and created hexagonally packed mesopores, resulting in high surface area and porosity. In addition, after establishing a suitable Cu^2+^ biological concentration that did not affect the survival and growth of fibroblasts, the authors demonstrated that the nanocellulose/Cu-MBG aerogels had an angiogenic effect and significantly up-regulated the angiogenic-related gene expression (Vegf, Fgf2, and Pdgf) of 3T3 fibroblasts. Furthermore, the Cu^2+^ released from the Cu-containing composites also inhibited the growth of *Escherichia coli*.

During chronic wound healing, one of the major factors that can hinder the re-epithelization process and the closure of the wound is the prolonged inflammation [179]. It is characterized by a remarkably high proteolytic activity, which has a great impact on cell proliferation as, during this stage, the growth factors and extracellular matrix tissue are degraded. Therefore, the development of materials that can detect the presence of high proteolytic activity at the wound site while promoting wound healing is in great search. In this regard, smart nanocellulose-based aerogels endowed with a short fluorescent peptide sequence were evaluated as transducer surfaces for biosensors and confirmed to have potential as sequestering biocompatible dressings, demonstrating high selectivity and sensitivity for protease in chronic wound fluid [173].

Lu et al. presented the preparation of an aerogel based on dialdehyde nanocellulose and collagen [161]. Due to the porosity and good biocompatibility, these materials are promising to be used in wound care applications [161,171,174].

Alginate, which is a polysaccharide that has been already applied in various biomedical applications, was utilized by Franco et al. [180] in synthesizing an innovative aerogel to promote wound closure. They obtained the matrix by supercritical impregnation of mesoglycan (MSG) onto calcium alginate aerogel. The newly obtained aerogel stimulated the re-epithelialization process and also acted as a barrier against wound infections.

Collagen, a protein with a major role in the wound healing process, has been also utilized in developing biocompatible aerogels as dressings. Curcumin cross-linked collagen aerogels with controlled anti-proteolytic activity and pro-angiogenic gene expression were synthesized [181]. Curcumin not only increases collagen stability, but it also imparts its intrinsic therapeutic properties such as antioxidant, anti-fungal, anti-viral, and anti-inflammatory properties, to the collagen matrix. The obtained curcumin/collagen-based aerogels presented a porous 3D network with uniform distributed pores, which closely mimicked the in vivo extracellular matrix (ECM) characteristic, making them suitable for wound care applications.

Another nutraceutical-reinforced collagen aerogel was developed by Govindarajan et al. through the embedment of wheat grass into a collagen matrix [182]. The inclusion of bioactive wheat grass not only enhanced the physicochemical and biomechanical properties of the aerogels, but also endowed the collagen-based aerogel with valuable therapeutic properties through its contained components, such as chlorophyll, vitamin E, and vitamin C and anti-anemic factors, such as vitamin B12, iron, folic acid, pyridoxine, amino acids, and enzymes.

## 6. Conclusions and Outlook

The unusual properties of aerogels, as well as their unique processing strategy, make them the most exciting materials with applicability in nanotechnology, even after incorporating a variety of nanomaterials into the aerogel matrix [185].

Various strategies were developed for the inclusion of high-performing nanomaterials and the improvement of aerogel-based composite systems. The new aerogels with incorporated nanomaterials present improved mechanical stability and strength and offer unique functionalities as high electrical conductivities, thermal stability, and reactivity.

In the context of vast domains of applicability of the aerogels as nanostructured materials presented in the reviews, the full potential of these materials is still to be assessed for various technology sectors. Thus, García-González’s review mentions that the use of aerogels in emerging applications, such as biotechnological applications related to environmental sciences and biomedical applications, should be further explored [186].

The present review aims to provide an overview of the obtaining and applicability of biobased aerogels. Due to their unique structures (porous 3D networks with high specificity, low density, low dielectric constant, and good mechanical properties), biobased aerogels have been extensively studied in the last few years. In addition, these aerogels offer a wider range of applicability and often improved performance than precursor renewable materials. As Kistler pointed out in 1931, aerogels can be obtained practically from a variety of raw materials, which explains the emergence of new aerogels with different applications every year. The flexibility of the obtaining conditions (variation of the parameters of the synthesis, composition, etc.) allows the controlled design of a versatile aerogel network that can be adapted to specific applications. Regarding, the methods of preparing biobased aerogels, they have been described in the review. As for the other aerogels, the drying stage is of major importance, being responsible for both maintaining porosity and maintaining integrity. Among the different drying methods, the supercritical drying technique is the most efficient method of attaining well-defined structures, while the freeze-drying is much easier, cheaper, and more environmentally-friendly. Renewable materials are the materials of the moment because they are cheap, non-toxic, and abundant, lowering the manufacturing costs of aerogels. At the end of the review, recent biomedical applications of biobased aerogels are presented, including drug administration, tissue engineering, and wound healing. Due to their adjustable chemical composition, special porosity, and good mechanical properties, the aerogels satisfy the requirements for use as a scaffold in tissue engineering. They also have found their applicability in the design of implantable cardiovascular devices. Moreover, due to their high porosity and large specific surface area, aerogels can be used as matrices for various biomolecules for detection applications. They are also successfully used as platforms for adsorption and controlled release of various bioactive compounds. Although it is known and increasingly mentioned, the potential of aerogels in the medical field is not sufficiently explored. For example, the use of aerogels in tissue engineering and wound care is not satisfactorily exploited, although it is evident that their properties make them indispensable in regenerative medicine.

Although it is clear that, by adapting the conditions of manufacturing and processing (especially drying), biobased versatile aerogels with multiple applications can be obtained, there is still a lot of research work to be undertaken and many directions to be followed to obtain biomaterials that can be marketed. Even though new types of aerogels have emerged lately, there is still a need to create new species of single-component or hybrid aerogels. Single component aerogels, although valuable, are restricted in applications by their uniqueness. For this reason, the development of hybrid aerogels that have intelligent, multiple functions is more useful in practice. In this category, aerogels can be formed from a renewable polymer and an inorganic material (e.g., carbon nanofibrils), but also by a natural polymer combined with a synthetic polymer.

## Figures and Tables

**Figure 1 pharmaceutics-12-00449-f001:**
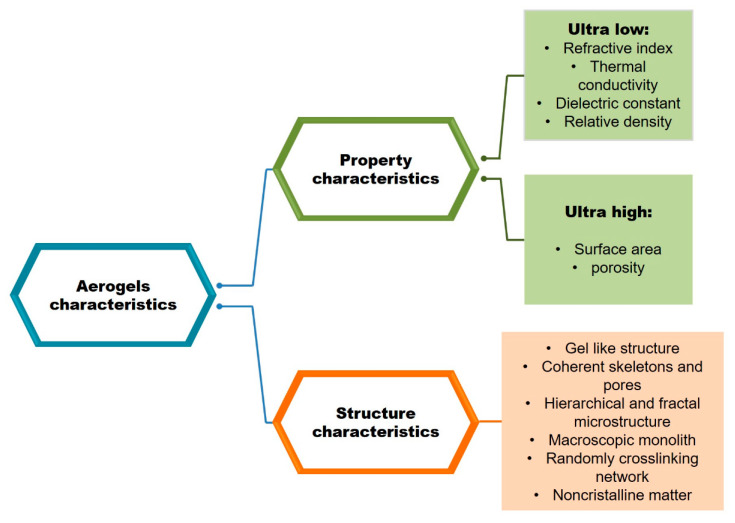
The main characteristics of aerogel type materials (adapted from Reference [24]).

**Figure 2 pharmaceutics-12-00449-f002:**
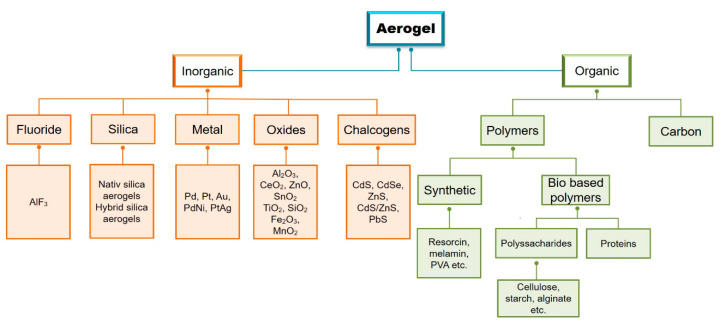
Classification of aerogels according to the nature of the materials used.

**Figure 3 pharmaceutics-12-00449-f003:**
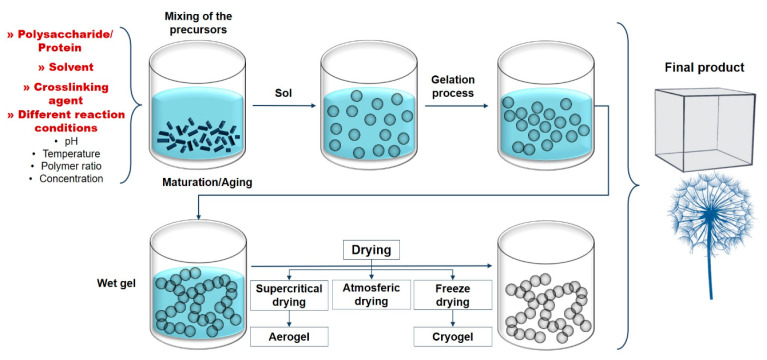
Methods for obtaining bio-based aerogels.

**Figure 4 pharmaceutics-12-00449-f004:**
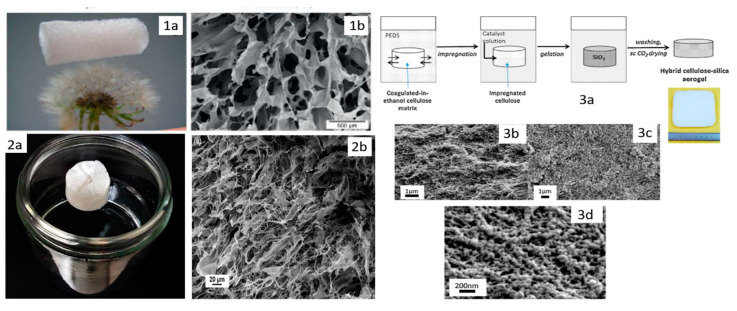
(**1**) TEMPO-oxidized and mechanically blended cellulose nanofibrils (CNFs): photograph of a 0.2 CNF aerogel on top of a dandelion (**1a**) and SEM images of 0.2 CNF aerogel pore structures (**1b**). Reprinted with permission from [56]. Copyright (2014) Royal Society of Chemistry. (**2**) Photograph of floating hydrophobic CNF aerogel on water surface (**2a**) and Field Emission Scanning Electron Microscopy (FE-SEM) images of CNF aerogels (**2b**). Reprinted with permission from [59]. Copyright (2016). (**3**) Schematic illustration of the preparation route of cellulose-silica composite aerogels, with a photograph of a composite aerogel sample (**3a**), SEM images of composite aerogels from 3% cellulose-1-ethyl-3-methylimidzolium acetate- (EmimAc) DMSO solution, manufactured with molecular diffusion (**3b**) and forced flow impregnation (**3c**,**d**). Reprinted with permission from [60]. Copyright (2015) Elsevier.

**Figure 5 pharmaceutics-12-00449-f005:**
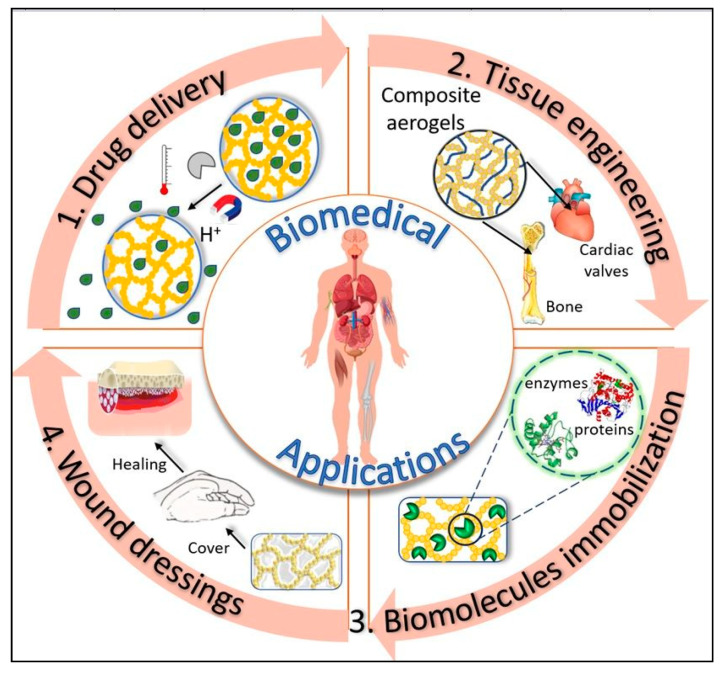
Applications of aerogels in the biomedical field.

**Figure 6 pharmaceutics-12-00449-f006:**
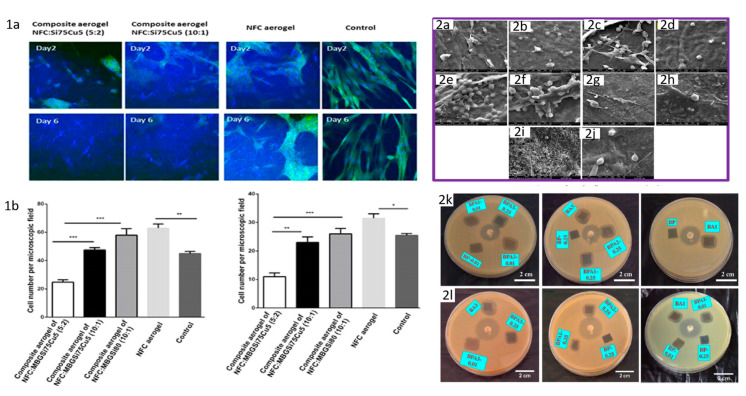
In vitro biological test results for several cellulose-based aerogels: (**1**) Representative confocal images of the expression of cytoskeleton marker actin (in green) and nucleus marker 4′,6-diamidino-2-phenylindole (DAPI) (in blue) for composite aerogels of NFC and mesoporous bioactive glass (MBG) incubated with 3T3 fibroblast cells (**1a**), and cell proliferation profiles measured after 2 days and 6 days of incubation, * *p* < 0.1; ** *p* < 0.01; *** *p* < 0.001 (**1b**). Reprinted with permission from [33]. Copyright (2016) Elsevier. (**2**) FE-SEM images of cells seeded on pristine bacterial cellulose (**2a**,**2b**) and bacterial cellulose/AgNPs/polyaniline (PANI): bacterial cellulose/PANI aerogel in 0.25 M HCl (BP-0.25) (**2c**,**2d**), bacterial cellulose/AgNPs aerogel (BA3) (**2e**, **2f**), bacterial cellulose/AgNPs/PANI aerogels in 0.01 M HCl (BPA3-0.01) (**2g**,**2h**), and in 0.25 M HCl (BPA3-0.25) (**2i**,**2j**), respectively, after 1 day (**2a**,**2c**,**2e**,**2g**,**2i**) and 7 days (**2b**,**2d**,**2f**,**2h**,**2j**); antibacterial activities of composite aerogels against *Staphylococcus aureus* (*S. aureus*) (**2l**) and against *Escherichia coli* (*E. coli*) (**2k**), respectively. Reprinted with permission from [172] Copyright (2020) Elsevier.

**Table 1 pharmaceutics-12-00449-t001:** Biobased aerogels.

Biobased Aerogels	References
Cellulose-based aerogels	[12,23,25,27,39,53,54,55,56,57,58,59,60,61]
Lignin-based aerogels	[62,63,64,65,66,67,68]
Pectin-based aerogels	[4,26,68,69,70,71]
Alginate-based aerogels	[3,13,72,73,74,75,76,77,78,79,80,81,82,83,84]
Starch-based aerogels	[85,86,87,88,89,90,91]
Chitosan-based aerogels	[12,92,93,94,95,96,97,98,99,100,101,102,103]
Protein-based aerogels	[7,26,104,105,106,107,108,109,110,111,112,113,114,115,116,117,118]

**Table 2 pharmaceutics-12-00449-t002:** Applications in the biomedical field.

Fields of Applications of Aerogels	References
**Aerogel in drug delivery**	[7,74,144,145,146,147]
chitosan	[98,100,102,148]
alginate	[75,133,149]
celulose	[150,151]
gelatin	[152]
pectine	[68,153,154]
protein	[104,143,155]
**Aerogel for tissue engineering**	[121,156,157]
collagen/alginate	[158]
chitin-hydroxyapatite composites	[159]
alginate-lignin	[160]
nanocellulose	[161]
silica	[162,163]
chitosan	[148]
**Aerogel for biomolecules immobilization**	[158,164,165]
**Aerogel for wound care**	[166,167,168,169,170]
cellulose	[171,172,173]
nanocellulose	[174,175,176,177]
chitosan	[178,179]
alginate	[180]
collagen	[181,182]

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
