# Peer review of "New Trends in Bio-Based Aerogels"

_pharmaceutics, 2020, doi:10.3390/pharmaceutics12050449_

Round 1

Reviewer 1 Report

Manuscript Number:  780857

New trends in bio-based aerogels

Reviewer(s)’ General Comments to Author:

The performed experimental techniques, obtained and discussed data in this paper should be published in the Pharmaceutics after minor revision.

Are used ionic liquids for aerogel manufacturing? I saw only 2 articles cited about ionic liquid, despite a lot of them and their expansion in the scientific community last decade. Please, add some investigations based on ionic liquids such as solvents or components, precursors , if such articles exist.

The manuscript could benefit from more careful proofreading to eliminate the errors due to poor word usage. The authors should be careful with the typographical things. Physical sizes should be written in italic form. Several errors have been found throughout the paper.Also, to improve English language.

Just one question, related to new trends in bio-based aerogels. My question is related to areas of application of aerogels. Could some aerogels be applied as some components in batteries?

The treatment of reliable data is correct and the obtained results are interesting. This paper acctually review , could be suitable for publication in this journal Pharmaceutics, while some improvements as mentioned in the letter above have been made.

Author Response

Answers to the Reviewers

  The manuscript, entitled:

New trends in bio-based aerogels

authors: Loredana Elena Nita, Alina Ghilan, Alina Gabriela Rusu, Iordana Neamtu and Aurica P. Chiriac

Firstly, we thank the Reviewers by helping us to improve our paper. The corrections were made and the manuscript was re-written accordingly with indications.

Comments from reviewers:

Reviewer 1

The performed experimental techniques, obtained and discussed data in this paper should be published in the Pharmaceutics after minor revision.

Are used ionic liquids for aerogel manufacturing? I saw only 2 articles cited about ionic liquid, despite a lot of them and their expansion in the scientific community last decade. Please, add some investigations based on ionic liquids such as solvents or components, precursors , if such articles exist.

  • As the reviewer have suggested, some articles that have as main goal preparation of bio-based aerogels using ionic liquids were introduced and a whole paragraph was written to explain in brief the importance of using such solvents in aerogel development.

<<As an alternative to the conventional organic solvents used as templates for porous materials, in the manufacturing of aerogels based on polysaccharides are frequently used ionic liquids (IL), also known as “green solvents”.  ILS are organic salts that are liquids at room temperature, with high thermostability and electric conductivity and are environment friendly exhibiting easy recyclability [28]. As the dissolution of some polysaccharides in organic solvent or water is the major disadvantage that can hinder their utilization in obtaining aerogels for medical applications, ILS can provide the most needed processing platforms of these biopolymers into high added-value materials. Moreover, the combination of green solvents, such as ILs, used in the dissolution of polysaccharides together with the utilization of an environmentally friendly manufacturing technique such as supercritical fluid technology (SCF) which can easily be adapted to extract the ILS, have enabled the development of various porous bio-based aerogels. Thus, polysaccharides like chitin [28], cellulose [29-31], starch [32] or lignin [33] have been used to obtain aerogels with different degree of porosity using ILS. Moreover, ILS like 1-allyl-3-methylimidazolium chloride (AMIMCl) contributes to the gelation of cellulose leading to formation of nanoporous materials with uniform structure and high degree of flexibility [31]. On the other hand, one of the physical properties of ILS that can influence the surface area of the materials is the melting point. Lopes and collaborators have highlighted in their study the influence of such property on preparing porous cellulose-based aerogels. They observed that when ILS with lower melting points like 1-ethyl-3-methylimidzolium acetate (EmimAc) or 1-ethyl-3-methylimidazolium diethyl phosphate (EmimDEP) were utilized, aerogels with higher surface areas were formed [34].>>

The manuscript could benefit from more careful proofreading to eliminate the errors due to poor word usage. The authors should be careful with the typographical things. Physical sizes should be written in italic form. Several errors have been found throughout the paper. Also, to improve English language.

We taken into consideration all reviewer observations: the manuscript was carefully checked, physical sizes was written in italic forms, English was improved.

Just one question, related to new trends in bio-based aerogels. My question is related to areas of application of aerogels. Could some aerogels be applied as some components in batteries?

  • Also, the question imposed by the reviewer regarding the utilization of bio-based aerogels as components of batteries have been answered.

The only utilization of bio-based aerogels with energy storage device functionality is as electrode materials for supercapacitors which can be employed together with batteries for emergency backup power sources and public transportation vehicles. But they have to have higher specific energy and therefore the materials will need to possess high specific surface area, high electronic and ionic conductivities and high porosity. These material properties are accomplished with success by some aerogels like the ones based on cellulose and regenerated by using 1-allyl-3-methylimidazolium chloride (AMIMCl) as ionic liquid. Moreover, one recent article discusses the utilization of carbon aerogels based on kraft and organosolv lignins as materials for nano-capacitors [Yang, B. S.; Kang, K.-Y.; Jeong, M.-J. Preparation of lignin-based carbon aerogels as biomaterials for nano-supercapacitor. Journal of the Korean Physical Society 2017, 71, 478–482].

The treatment of reliable data is correct and the obtained results are interesting. This paper actually review , could be suitable for publication in this journal Pharmaceutics, while some improvements as mentioned in the letter above have been made.

The authors want to thank to the Reviewers and to Editor or the useful and pertinent suggestions which gives the chance to improve the work.

                                                                                                          Aurica P. Chiriac

Reviewer 2 Report

This review paper provides a full update on bio-based aerogels. It lists both the systems looked at so far and their uses. It represents a real interest, however several points are to be modified and / or supplemented:
1- In Figure 1 it is mentioned among the physical properties an important refractive index for aerogels, but it is commonly accepted that the values ​​of the refractive indices for these materials are low.
2- Figure 2 must be completed and modified, by clearly specifying the oxides (silica, zirconia, titanium oxide, YAG ...), metals (gold ...), chalcogenides (CdS, CdSe ...) and fluorides (GdF3 ...). It would be useful to mention the work of German, Swiss and French teams in this field of inorganic aerogels.
3- Why do xerogels appear in Figure 3. It is well known that this type of material is not aerogel.
4- p4: it is important to clearly specify the drying methods, their conditions of implementation, the structures obtained and to compare them with supercritical drying.
5- p4; line 112: "After the release of the gas" instead of "After the release of the liquid" ...
6- p4, line 114: "interface" instead of "meniscus"
7- p4: give the critical temperature and the critical pressure of the solvents.
8- p4, line 122: In Berglund's article supercritical drying is not used. He uses sublimation of the solvent.
9- p5, line 148: The experimental conditions must be specified, otherwise a xerogel can be obtained!
10- p5, line 154: "eliminated" instead of "dried"
11- p5, line 164: Another drawback is the change in volume when the water is frozen.
12- p5, line 168: Is it really freeze-drying? Give a reference.
13- p5: overall the freeze-drying part is a catch-all part, the authors must review this part to clarify things!
14- p10: It is necessary to lighten the style to facilitate reading and understanding. Line 386 to line 391: only one sentence ...
15- p10-11, line 433-436: same remark.
16- p13, line 561: the sentence is not finished ...
17- p16, line 706: irradiated with what?
18- Generally, explanatory figures are missing, more are needed.
20- Repeat a thorough reading to correct typing errors and correct English.

Author Response

Answers to the Reviewers

  The manuscript, entitled:

New trends in bio-based aerogels

authors: Loredana Elena Nita, Alina Ghilan, Alina Gabriela Rusu, Iordana Neamtu and Aurica P. Chiriac

Firstly, we thank the Reviewers by helping us to improve our paper. The corrections were made and the manuscript was re-written accordingly with indications.

Comments from reviewers:

Reviewer 2

This review paper provides a full update on bio-based aerogels. It lists both the systems looked at so far and their uses. It represents a real interest, however several points are to be modified and / or supplemented:

1- In Figure 1 it is mentioned among the physical properties an important refractive index for aerogels, but it is commonly accepted that the values ​​of the refractive indices for these materials are low.

The Figure 1 was corrected.

2- Figure 2 must be completed and modified, by clearly specifying the oxides (silica, zirconia, titanium oxide, YAG ...), metals (gold ...), chalcogenides (CdS, CdSe ...) and fluorides (GdF3 ...). It would be useful to mention the work of German, Swiss and French teams in this field of inorganic aerogels.

The Figure 2 was corrected.

3- Why do xerogels appear in Figure 3. It is well known that this type of material is not aerogel.

The Figure 3 was corrected.

4- p4: it is important to clearly specify the drying methods, their conditions of implementation, the structures obtained and to compare them with supercritical drying.

Drying methods have been restructured to be clearer.

5- p4; line 112: "After the release of the gas" instead of "After the release of the liquid" ...

The modification was made.

6- p4, line 114: "interface" instead of "meniscus"

The modification was made.

7- p4: give the critical temperature and the critical pressure of the solvents.

The following paragraph was inserted in text:

<<Supercritical drying requires specific conditions that differ depending on the solvent used. Thus, for example when the solvent is water the required Tc is about 374 oC and Pc about 22 Pa. Contrary, ethanol requires a Pc about 6.3 Pa and Tc = 243 oC [33b].>>

8- p4, line 122: In Berglund's article supercritical drying is not used. He uses sublimation of the solvent.

The phrase was modified:

<<Berglund used sublimation instead sc by producing an aqueous dispersion of cellulose, solvent exchanged further with tert-butanol.>>               

9- p5, line 148: The experimental conditions must be specified, otherwise a xerogel can be obtained!

The following paragraph was inserted in text:

<< Thus, organic aerogels were obtained using a sol-gel process followed by solvent exchange with tipical solvent (such us acetone, ethanol) and then dried under ambient pressure condition.>>

10- p5, line 154: "eliminated" instead of "dried"

The modification was made.

11- p5, line 164: Another drawback is the change in volume when the water is frozen.

The modification was made.

12- p5, line 168: Is it really freeze-drying? Give a reference.

The following sentences was introduced before line 168:

<<II.4. Other methods:>>

13- p5: overall the freeze-drying part is a catch-all part, the authors must review this part to clarify things!

This part was revised.

14- p10: It is necessary to lighten the style to facilitate reading and understanding. Line 386 to line 391: only one sentence ...

The paragraph was rewritten:

<<Solvents used as well as the drying method of the gel are key parameters that influence the final characteristics of alginate aerogels. Thus, Rosalía Rodríguez Dorado [70] investigated xerogel, cryogel or aerogel gel beads obtained from alginates of different weights and molecular concentrations and using different gelation conditions and drying methods (eg supercritical drying, frozen drying and oven drying). The study highlights the stability of the physico-chemical properties of alginate aerogels in interdependence with the obtaining and storage conditions.>>

15- p10-11, line 433-436: same remark.

The paragraph was rewritten:

In this model, the porous structure of the aerogels was represented at the mesoscale level as a set of solid particles connected by solid bonds. Thus, an elastic-plastic functional model is developed to describe the rheological behavior of alginate aerogels with varying degrees of cross-linking, namely calcium content.

16- p13, line 561: the sentence is not finished ...

The sentence was finished.

17- p16, line 706: irradiated with what?

These are the only information’s give in the paper [70]:

the obtaining of a homogeneous solution, the samples were irradiated using 400 W for 20 min, and freeze dried resulting in a very porous material

18- Generally, explanatory figures are missing, more are needed.

2 new Figures and 2 new Tables was added in the paper.

20- Repeat a thorough reading to correct typing errors and correct English.

The manuscript was carefully check and English was improved.

The authors want to thank to the Reviewers and to Editor or the useful and pertinent suggestions which gives the chance to improve the work.

                                                                                                          Aurica P. Chiriac

Reviewer 3 Report

This is an interesting paper reviewing aerogels.  Although this is becoming a very broad area the authors do well in touching on some of the highlights of the promise, preparation, and uses of aerogels.  Paper is generally well written, but still has several places that need correcting.

Abstract is full of grammatical errors.

LIne: 73-77 should break into more than one sentence. The whole paragraph is one sentence.

Line: 78 should have a new subtitle, and in addition the authors should make more use of subtitles and organizing broader themes they present and sub-ordering into smaller themes.  Line 92-98 is another example of this. The paper goes from I. Introduction to II Drying....

Line 113: Freeze drying tends to collapse aerogels. However this is not pointed out.

Line 224: Needs to start new paragraph and subtitle on cellulose aerogels.

LIne 446: Alginate applications should be elaborated on separately.

Line 506: Regernerative medicine applications should be elaborated on rather than an after thought.

LIne 600: 'In its structure are amino acids' should be deleted. Casein is a protein and thus it is understood to be composed of amino acids.

Line 621-654:  There is a lot of biomedical research going on with aerogels.  The authors should be more careful in summarizing it.  Suggest  taking comments buried  in the manuscript on biomedical and putting them in one place or lengthen this section, or putting this section in the introduction.

Line 778: Nanocellulose should be treated as a separate subtitled area.  This is a very prolific area with applications in important biomedical and sensor applications and should not be overlooked in a pharmaceutics journal.

LIne 832:  Aerogels for woundcare has not been treated sufficiently.  For example see Edwards, Fontenot, Liebner on nanocellulose aerogels for chronic wounds.

Lines 879-907: Paragraph is too long for a conclusion.

Author Response

Answers to the Reviewers

  The manuscript, entitled:

New trends in bio-based aerogels

authors: Loredana Elena Nita, Alina Ghilan, Alina Gabriela Rusu, Iordana Neamtu and Aurica P. Chiriac

Firstly, we thank the Reviewers by helping us to improve our paper. The corrections were made and the manuscript was re-written accordingly with indications.

Comments from reviewers:

Reviewer 3

This is an interesting paper reviewing aerogels.  Although this is becoming a very broad area the authors do well in touching on some of the highlights of the promise, preparation, and uses of aerogels.  Paper is generally well written, but still has several places that need correcting.

Abstract is full of grammatical errors.

Abstract was rewritten.

LIne: 73-77 should break into more than one sentence. The whole paragraph is one sentence.

The paragraph was rewritten:

<<Although in the last period a lot of reviews have been published on the topic of obtaining and applicability of aerogels, this type of literature is beneficial, given the multitude of articles on this topic that appear at an extremely intense rhythm, but also due to the innovative aspects that underlie the obtaining of these materials.>>

Line: 78 should have a new subtitle, and in addition the authors should make more use of subtitles and organizing broader themes they present and sub-ordering into smaller themes.  Line 92-98 is another example of this. The paper goes from I. Introduction to II Drying....

Other subtitle was inserted in text such us: ,<< II. Methods of preparation:>>

Line 113: Freeze drying tends to collapse aerogels. However this is not pointed out.

The following paragraph was inserted in text:

<<Disadvantages of this method are the long processing time, the change in volume when the water is frozen which sometimes can produced collapse of aerogels and also, high energetically consumption.>>

Line 224: Needs to start new paragraph and subtitle on cellulose aerogels.

A new paragraph was started and the subtitle:<< IV.1.1 Type of cellulose>> was inserted

LIne 446: Alginate applications should be elaborated on separately.

The alginate applications was discussed in the V parts, Biomedical applications of aerogels. In each biomedical application, alginate-based aerogels were mentioned, with the biological properties suitable for each application.

Line 506: Regernerative medicine applications should be elaborated on rather than an after thought.

The fragment was rewritten.

LIne 600: 'In its structure are amino acids' should be deleted. Casein is a protein and thus it is understood to be composed of amino acids.

The sentence was removed.

Line 621-654:  There is a lot of biomedical research going on with aerogels.  The authors should be more careful in summarizing it.  Suggest  taking comments buried  in the manuscript on biomedical and putting them in one place or lengthen this section, or putting this section in the introduction.

The observation was taken into consideration.

Line 778: Nanocellulose should be treated as a separate subtitled area.  This is a very prolific area with applications in important biomedical and sensor applications and should not be overlooked in a pharmaceutics journal.

The observation was taken into consideration.

LIne 832:  Aerogels for woundcare has not been treated sufficiently.  For example see Edwards, Fontenot, Liebner on nanocellulose aerogels for chronic wounds.

The following paragraphs were inserted:

<<Among them, chitin and chitosan-based aerogels occupy a special place due to their special properties (antifungal and bactericidal character, high permeability to oxygen), promoting the fibroblast proliferation). In order to treat chronic wounds and to prevent the subsequent contamination with pathogen agents, vancomycin, an antimicrobial drug, was loaded into chitosan aerogel beads manufactured through scCO2 technique. The high porosity (> 96%) and the large surface area (> 200 m2/g) promoted a fast release of the drug, highly necessary at the wound site  in order to prevent the spreading of microbes shortly after debridement. In addition, the in vitro cytocompatibility on fibroblasts evidenced no harmful effect on cells, highlighting the potential of these chitosan-based aerogels as dressings in management of chronic wounds [175]].

      Another polysaccharide successfully used as a wound dressing is bacterial nanocellulose that is non-toxic, non-allergic and biocompatible. Moreover, nanocellulose can endow the dressings with the capacity to absorb and retain moisture at the wound site, while having many advantageous effects in the wound healing process, like minimizing inflammatory response and stimulating fibroblast proliferation [176, 177]. Although bacterial cellulosed-based aerogels posses many unique features like nontoxicity nature, high porosity and excellent mechanical properties, they lack antimicrobial properties and stimuli-sensitive character which limit their utilization in wound care area. Through the embedment in their 3D network of inorganic nanoparticles or conductive agents, those properties were improved. In this regard, Hosseini and colleagues synthesized ternary nanocomposite aerogels based on bacterial cellulose embedding not only silver nanoparticles (AgNP) as antimicrobial agents, but also polyaniline (PANI) particles with rose-like morphology formed in-situ within aerogel network for tailored porosity. All aerogels had higher porosity (>80%) and great elastic properties suitable for a wound dressing. Although the cell attachment was not promoted onto nanocomposite aerogels, the cell proliferation recorded an on-going increase throughout the incubation period [178]. Another versatile composite aerogel based on nanofibrillated cellulose with copper-containing mesoporous bioactive glass (Cu-MBG) inclusions was successfully obtained by Wang and colleagues [179]. The reinforcement of the nanocellulose matrix with Cu-MBG particles enhanced the aerogels capacity of water absorption and created hexagonally packed mesopores resulting in high surface area and porosity. Also, after establishing a suitable Cu2+ biological concentration which did not affect the survival and growth of fibroblasts, the authors demonstrated that the nanocellulose/Cu-MBG aerogels had an angiogenic effect and significantly up-regulated the angiogenic-related gene expression (Vegf, Fgf2 and Pdgf) of 3T3 fibroblasts. Furthermore, the Cu2+ released from the Cu-containing composites also inhibited the growth of Escherichia coli.

During chronic wounds healing, one of the major factors that can hinder the re-epithelization process and the closure of the wound, is the prolonged inflammation [180]. It is characterized by a remarkably high proteolytic activity which has a great impact on cell proliferation as during this stage the growth factors and extracellular matrix tissue are degraded. Therefore, the development of materials which have the ability to detect the presence of a high proteolytic activity at the wound site while promoting the wound healing, are in great search. In this regard, smart nanocellulose-based aerogels endowed with a short fluorescent peptide sequence were evaluated as transducer surfaces for biosensors and confirmed to have potential as sequestering biocompatible dressings demonstrating high selectivity and sensitivity for protease in chronic wound fluid [181].

Lu and collaborators presented the preparation of an aerogel based on dialdehyde nanocellulose and collagen [161]. Due to the porosity and good biocompatibility these materials are promising to be used in wound care applications [161, 173,174]. 

Alginate which is a polysaccharide that has been already applied in various biomedical applications, was utilized by Franco and colleague [182] in synthesizing an innovative aerogel to promote wound closure. They obtained the matrix by supercritical impregnation of mesoglycan (MSG) onto calcium alginate aerogel. The newly obtained aerogel stimulated the re-epithelialization process and also acted as a barrier against wound infections.

Collagen, a protein with major role in wound healing process, has been also utilized in developing biocompatible aerogels as dressings. Curcumin cross-linked collagen aerogel system with controlled anti-proteolytic activity and pro-angiogenic gene expression were synthesized [183]. Curcumin not only increases collagen stability, but it also imparts its intrinsic therapeutic properties like anti-oxidant, anti-fungal, anti-viral and anti-inflammatory to collagen matrix. The obtained curcumin/collagen-based aerogels presented a porous 3D network with uniform distributed pores which closely mimicked the in vivo extracellular matrix (ECM) characteristic, making them suitable for wound care applications.

Another nutraceutical reinforced collagen aerogels was developed by Govindarajan and colleagues though the embedment of wheat grass into a collagen matrix [184].  The inclusion of bioactive wheat grass not only enhanced the physicochemical and bio-mechanical properties of the aerogels, but also endowed the collagen-based aerogel with valuable therapeutic properties through the its contained components such as  chlorophyll, vitamin E, vitamin C, anti-anemic factors like vitamin B12, iron, folic acid, pyridoxine, amino acids and enzymes>>

Lines 879-907: Paragraph is too long for a conclusion.

The paragraph was shortening.

The authors want to thank to the Reviewers and to Editor or the useful and pertinent suggestions which gives the chance to improve the work.

                                                                                                          Aurica P. Chiriac
